# Are Plant–Soil Feedbacks Caused by Many Weak Microbial Interactions?

**DOI:** 10.3390/biology12111374

**Published:** 2023-10-27

**Authors:** Julia K. Aaronson, Andrew Kulmatiski, Leslie E. Forero, Josephine Grenzer, Jeanette M. Norton

**Affiliations:** 1Wildland Resources and the Ecology Center, Utah State University, Logan, UT 84322, USA; jaaronson006@gmail.com (J.K.A.); leslieeforero@gmail.com (L.E.F.); josephinegrenzer@gmail.com (J.G.); 2Plants, Soils and Climate Department and the Ecology Center, Utah State University, Logan, UT 84322, USA; jeanette.norton@usu.edu

**Keywords:** field experiment, microbial community, plant–microbe interactions

## Abstract

**Simple Summary:**

Soil organism communities are typically different under different plant species. Often, soil bacteria and fungi that feed on a plant species will accumulate and decrease the growth of the dominant species. This effect has long been recognized in agricultural systems and more recently in more diverse and natural systems. Yet, little is known about the soil organisms that cause these effects. Here, we described the soil organisms in common garden experiments in North America and Europe and found that soil organisms vary widely among sites, among years, and between bulk soils and the soils around roots. Plant effects on the soil microbial community were only detected in root-associated soils. In these soils, plants changed dozens to hundreds of soil organisms. In short, plants affected a small but diverse portion of the soil microbial community, and these changes were found to affect subsequent plant growth. The results suggest that microbial-based management of plant growth will likely have to manipulate a small but diverse subset of the soil microbial community and not individual plant pathogens or symbionts.

**Abstract:**

We used high-throughput sequencing and multivariate analyses to describe soil microbial community composition in two four-year field plant–soil feedback (PSF) experiments in Minnesota, USA and Jena, Germany. In descending order of variation explained, microbial community composition differed between the two study sites, among years, between bulk and rhizosphere soils, and among rhizosphere soils cultivated by different plant species. To try to identify soil organisms or communities that may cause PSF, we correlated plant growth responses with the microbial community composition associated with different plants. We found that plant biomass was correlated with values on two multivariate axes. These multivariate axes weighted dozens of soil organisms, suggesting that PSF was not caused by individual pathogens or symbionts but instead was caused by ‘many weak’ plant–microbe interactions. Taken together, the results suggest that PSFs result from complex interactions that occur within the context of a much larger soil microbial community whose composition is determined by factors associated with ‘site’ or year, such as soil pH, soil type, and weather. The results suggest that PSFs may be highly variable and difficult to reproduce because they result from complex interactions that occur in the context of a larger soil microbial community.

## 1. Introduction

Plant–soil feedbacks (PSFs) have gained attention due to their potential to explain plant growth and coexistence [1,2,3]. PSFs are typically measured using a two-phase approach in which target plant species are grown in a common soil in Phase 1, and then, in Phase II, each species is grown on ‘self’ or ‘other’ soils (i.e., ‘self’ soils are cultivated by the same species and ‘other’ soils are cultivated by different plant species) [4,5,6]. This bioassay approach uses plant growth to measure the net legacy effect of previous plant growth. While plants can affect soil chemistry and soil structure, interactions with soil microbes are typically assumed to be the primary determinant of PSF [7,8,9]. Plants can change the soil microbial community through root production and exudation that encourages the growth of some microbes (e.g., mycorrhizae) or suppresses the growth of other microbes (e.g., soil pathogens) [3,4,7,10]. Similarly, the chemical composition of leaf and root litter can induce changes in the saprophytic microbial community [3].

The bioassay approach has provided good insight into the net effects of plant–soil microbial interactions, but several aspects of the existing PSF research limit understanding of the soil organisms that cause PSF [10]. Most PSF research uses short-term greenhouse experiments. These experiments have been found to produce PSF values that are not correlated with measurements conducted in field conditions [11]. Perhaps more importantly, most PSF research has measured plant growth but not soil microbial composition [12,13,14]. A growing number of studies address this knowledge gap by describing soil microbial communities on soils with different plant growth histories [3,15]. However, few studies have taken the next step of correlating those changes in the soil microbial community with subsequent plant growth [3,16,17]. This is an important next step because correlations between the soil community composition and subsequent plant growth have the potential to identify putative plant-growth-promoting and plant-growth-suppressing soil organisms [17,18]. These correlations would not prove causation, but they would help to prioritize the soil organisms that are likely to cause plant growth responses.

Given the lack of microbial community data in PSF experiments, it is not clear, for example, if PSFs are caused by strong interactions with one or a few soil organisms, or, alternatively, if PSFs are caused by many weak interactions [17,19]. A negative PSF could occur due to the accumulation of a species-specific plant pathogen (i.e., a strong negative interaction) [17,20]. Alternatively, a negative PSF could occur due to changes in dozens or hundreds of soil organisms [19]. For example, a plant could create a soil with few mycorrhizae and plant-growth-promoting bacteria, many generalist plant pathotrophs, and a community of bacteria and fungi that provide slow nutrient cycling [21,22,23].

The ‘few strong’ or ‘many weak’ hypotheses have important implications for plant management [18]. If plant growth responses are determined by a few strong interactions, then there is great potential to identify plant-growth-promoting or plant-growth-suppressing soil organisms that could be applied to manage plant growth [24]. However, if plant growth responses are caused by many weak interactions, then plant management approaches that can shift communities of soil organisms will be needed [25,26,27]. These hypotheses are not mutually exclusive. Some plants may be strongly affected by a single pathogen, while others are affected by many weak interactions. Further, some plants and some soils may be more susceptible to disease (e.g., ruderal species) [28]. Agricultural systems, for example, may be more susceptible to disease outbreak because the crop species provides an abundant food source and because the low-diversity soil microbial communities in agricultural systems are more susceptible to large variations in community composition [29]. Thus, it is important to test for ‘few strong’ vs. ‘many weak’ interactions across many species.

Whether caused by few or many interactions, PSF occurs in the context of large, diverse soil microbial communities whose compositions can vary widely in space and time [30,31,32]. There are many potential factors that determine microbial community composition. Studies have found that the main factors determining microbial communities are related to soil physiochemical characteristics, especially pH, but also include factors such as soil texture and nutrient and carbon availability [14,33,34]. There is less support for the roles of latitude and elevation on microbial diversity [32,34,35]. Microbial composition also varies over time, both within and between seasons [11,36,37]. Soil communities can also vary widely in small spaces within soils, for example, in the rhizosphere vs. bulk soils [38]. Finally, plant species themselves can alter microbial composition both directly and indirectly [39,40,41]. There are few comprehensive studies investigating the patterns of microbial community assembly through large spatial scales or long time scales, so the factors driving these patterns are still poorly understood [33,37]. There is evidence, however, for the microbial biodiversity–function relationship [32]. There remains, therefore, a need to improve understanding of how plants structure microbial communities relative to other drivers [15].

The goal of this study was to describe the microbial communities in two separate PSF field experiments. In each of these experiments, plants were grown in randomly assigned plots in a common garden for two years. The plants were then removed and replanted on either ‘self’ or ‘other’ soil. Aboveground plant biomass was harvested in these experiments. The plant biomass was, on average, ~25% different on ‘self’ than ‘other’ soil [42,43]. These experiments, therefore, demonstrated that the plants created soil conditions in two years in the field that changed the subsequent plant growth. These plant growth responses have been reported elsewhere [42,43]. Here, we describe the soil microbial communities in the soils in these two experiments. Because plants demonstrated PSF effects in these experiments, we expected that plant effects on microbial communities would emerge across the four years of the study and be largest in rhizosphere soils where plants have more immediate effects [44,45,46]. Finally, and perhaps most importantly, we were able to test whether the changes plants caused in microbial community composition were correlated with plant biomass. Positive correlations would suggest that a plant created a soil microbial community with plant-growth-promoting organisms. Negative correlations would suggest that a plant created a soil microbial community with plant-growth-suppressing organisms. No correlations suggest that plants cause many changes in the soil microbial community that do not feed back to affect subsequent plant growth. For example, plants may cause changes in saprophytic bacterial communities, but these changes in community composition may not change soil nutrient cycling (i.e., function).

Our objectives were
To compare the effects of site, year, soil location (bulk vs. rhizosphere), and plant identify on soil microbial composition.To test for a correlation between the microbial composition associated with different species and subsequent plant growth.

## 2. Methods

### 2.1. Study Sites

This study describes the soil microbial communities in two fully factorial, four-year-long PSF experiments performed at the Cedar Creek Ecosystem Science Reserve Long Term Ecological Research Site, East Bethel, MN, USA (45.403290 N, 93.1874411 W) and the Jena Experimental field site, Jena, Germany (50.951276 N, 11.620545 E). These experiments used the same experimental design, but different plant species at each site. PSFs were measured for sixteen species at Cedar Creek and a different nine species at Jena. Species were selected to provide insight into diversity–productivity experiments previously performed at each site [10,11]. Methods, site conditions, and PSF values for those studies are reported elsewhere [10,11]. Briefly, the Cedar Creek site is located on sandy soils (regosols) in the Nymore series (mixed, frigid, Typic Udipsamment) with pH values of ~5.7 and carbon contents of ~7 g C kg^−1^ soil [47]. During the four years of the study (2015–2018), the mean annual precipitation (MAP) was 723.0 mm and mean annual temperature (MAT) was 6.5 °C, which is consistent with 1963–2019 records at the site (769.3 mm and 6.6 °C). The Jena site is located on alluvial soils (eutric fluvisols) with a pH value range of 7.1 to 8.4 and organic carbon content of 5–33 g C kg^−1^ [48]. During the four years of the study (2015–2018), the MAP was 499.0 mm and the MAT was 10.4 °C, which is slightly warmer and drier than the long-term averages at the site for 2002–2018 (544 mm and 9.8 °C, respectively). The first and final years of the experiment (2015 and 2018) were drier than average, with 459 mm and 395 mm of precipitation, while 2017 was wetter than average (615 mm).

### 2.2. Experimental Design

At Cedar Creek, in Year 1 (2015), 10 g m^−2^ of live seed was planted in 170 replicate plots for each of the 16 target species (Table 1). At the end of the growing season in Year 2, aboveground and belowground biomass was removed and each of the 16 species was planted on thirty-five replicate plots with ‘self’ soil and nine replicate plots of soil cultivated by each of the other 15 species used in the experiment. During Year 3, plots were weeded of non-target species and target plants allowed to grow. At the end of Year 4, aboveground plant biomass was clipped, dried, and weighed.

At Jena, nine plant species (Table 2) that had been previously grown at the site were randomly assigned to 139 common garden plots and grown for two years before being killed with herbicide and replanted with either the same plant species or other plant species in the experiment. In Year 1, 4 g/m^2^ of live seed was planted in 1251 plots. Due to poor establishment, *Anthriscus sylvestris* and *Geranium pratense* were reseeded in the fall of Year 1. At the end of the growing season in Year 2, above- and belowground biomass was removed. Each of the nine species was planted on 14 replicate plots of self-cultivated soils at the beginning of the growing season in Year 3. The aboveground biomass was clipped, dried, and weighed at the end of the growing season in Years 3 and 4.

### 2.3. Soil Sampling

At the end of each growing season (September), three replicate bulk soil samples were collected from self-cultivated plots at each site. In addition, a rhizosphere sample was collected at the end of the experiment when it was possible to destructively harvest plants. This design allowed us to examine microbial community composition between sites, over time, between bulk and rhizosphere samples, and among soils cultivated by different plant species. All plots contained vegetation; samples were not collected from ‘bare soils’. At Cedar Creek, a single 10 cm × 2.5 cm soil core was taken from three randomly selected plots for each of the self-cultivated soils for microbial analysis. In Jena, a single soil core from three replicate plots was combined to create one of three replicate samples for each ‘self’ soil. The soil samples were immediately put on ice and placed in −80 °C storage within a few hours of sampling. In the final year of the study, rhizosphere soil was collected from three mature plants in each of the three replicate plots for each plant species. Excavated plant roots were placed into a sterile plastic bag and shaken in the bag for 2 min. Soil samples were stored in the −80 °C freezer for one week for samples from 2015 to 2017 and for one month for samples from 2018, prior to DNA extraction.

### 2.4. DNA Extraction

DNA was extracted from the bulk and rhizosphere soil samples using a PowerSoil DNA Isolation Kit (MO BIO Laboratories, Inc., Carlsbad, CA, USA). DNA concentrations were checked using PicoGreen assay on a Modulus Microplate reader (Turner BioSystems, Inc., San Jose, CA, USA). Purified DNA was diluted to a maximum concentration of 6.0 ng/μL for bulk soil samples and 50.0 ng/μL for rhizosphere soil samples. All samples were stored at −80 °C until sequencing. Library preparation and sequencing were performed at Argonne National Laboratories Biosciences Division using the Earth Microbiome Project protocols for 16S using primer set 515F-806R for bacteria and archaea [49,50] and for ITS1 target using primers ITS1f–ITS2 for fungi [51]. The amplicons were sequenced on an Illumina MiSeq platform (Novogene Corporation, Beijing, China) using the Earth Microbiome Protocol. Sequences were then processed using the QIIME2 pipeline [52]. The DADA2 package was used to denoise samples and remove chimeras [51]. The forward and reverse reads were joined before denoising for the bacterial data. For the fungal data, only the forward reads were used due to the poor quality of the reverse reads (Phred quality score less than 13). For both fungal and bacterial samples, reads with expected errors higher than 1 were removed and truncated when quality was less than 13. Four samples were removed due to low sequence count (<50 sequences). Amplicon sequence variants (ASVs) were used for the downstream analysis, so no clustering was performed. Taxonomy was assigned using the Silva and UNITE reference databases [50,52] and a naïve Bayesian classifier [51,53]. In total, 260 bacterial and 1843 fungal amplicon sequence variants (ASVs) were obtained for both sites.

### 2.5. Statistical Analyses

Statistical analyses were computed using R v4.2.2. [54]. Total sum scaling (TSS) was used on the raw count data obtained from sequencing. NMDS based on the Bray–Curtis dissimilarity from the package “vegan” [55] was used to examine the overall patterns in the microbial communities. Detection of site, year, soil location, and plant effects on the structure of the microbial communities was conducted using separate PERMANOVAs, also from the vegan package. Significance was evaluated at a *p*-value of 0.05 and the intensity of the signal was measured using the R-squared (R^2^) value. Envfit from vegan was used to fit ASV vectors onto the ordination, with p adjustment using the Benjamini–Hochberg method to control for the false discovery rate. Differential abundance was calculated using ANCOMBC2 [56]. Fungal trophic modes were assigned using the FUNGuild Database [57]. Trophic modes were only able to be obtained for ASVs identified to the family level. Linear regression (lm function) was performed to assess the effect of microbial community on plant biomass (log-transformed). The logged biomass of a species in a plot and the NMDS score of the microbial community in that plot represent the y and x coordinates of that point. Regression was only performed for Cedar Creek data as the rhizosphere soil in Jena was sampled from different plots than the plant biomass data.

## 3. Results

### 3.1. Microbial Community Composition

Sequencing of the 16S amplicon yielded a total of 4,718,857 high-quality non-chimeric sequences across all samples (including both sites), with a median of 14,792 sequences per sample (range 1171–31,674 sequences per sample). Sequencing of the ITS2 region yielded 5,294,116 high-quality fungal sequences with a median of 16,390 sequences per sample for both sites (range 2734–32,972 sequences per sample). Sequence Read Archive (SRA) has accession number PRJNA683074 and is available at www.ncbi.nlm.nih.gov/bioproject/683074 (accessed on February 2023). Of the 375 soil samples in our experimental design, 323 produced high-quality reads.

There were 260 bacterial and 1843 fungal ASVs identified in the dataset from 24 phyla and 355 families. There were 1370 fungal and 232 bacterial ASVs in Cedar Creek and 959 fungal and 194 bacterial ASVs in Jena. The most abundant ASVs (relative abundance > 1%) were found in association with almost every plant species and represent predominantly fungal taxa from *Ascomycota*, *Basidiomycota*, *Mortierellomycota*, and bacterial taxa from *Firmicutes*, *Proteobacteria*, and *Bacteroidetes*. These abundant ASVs represented 79.82% (Cedar Creek) and 75.79% (Jena) of all sequences, numbers typical of other studies [15]. Fungal (18.01%, 15.07%) and bacterial (12.00%, 14.89%) ASVs that were unable to be assigned to a phylum made up a significant portion of the microbial communities at Cedar Creek and Jena, respectively.

### 3.2. Effects of Site, Year, and Soil Location

There were averages of 122 ASVs per sample in Cedar Creek and 162 ASVs per sample in Jena. Ordination revealed that the two sites harbored distinct microbial communities (Figure 1), with site explaining 14.15% of the variance (*p* = 0.001). While distinct, the sites also shared 610 ASVs (29.03%). The relative abundance of bacterial phyla at both sites was similar, with *Firmicutes* dominating, followed by *Bacteroidetes* and then *Proteobacteria*. ANCOM revealed that 13 of 19 phyla differed between the two sites (Figure 2): *Mortierellomycota*, *Rozellomycota*, *Zoopagomycota*, *Kickxellomycota*, *Olpidiomycota*, and *Actinobacteria* had greater relative abundance in Jena than in Cedar Creek. The relative abundance of *Glomeromycota*, *Basidiomycota*, *Ascomycota*, *Monoblepharomycota*, *Mucoromycota*, *Proteobacteria*, and *Tenericutes* was greater in Cedar Creek than Jena.

In Cedar Creek, year explained 16.25% of the variance (*p* < 0.001). In Jena, year explained 22.14% of the variance (*p* < 0.001). *Firmicutes* and *Proteobacteria* dominated the bacteria in Cedar Creek, and *Firmicutes* and *Bacteroidetes* dominated the bacteria in Jena, with a high relative abundance of *Proteobacteria* in the first year of the study (Figure 3). Both sites showed a decrease in *Bacteroidetes* in the second year following disturbance (2016 and 2018). Additionally, Cedar Creek showed an increase in *Ascomycota*, *Glomeromycota*, and *Proteobacteria* and a decrease in *Monoblepharomycota* and *Mortierellomycota* in the second year following disturbance. For the fungi assigned fungal guilds, both sites showed an increase in pathotroph–saprotrophs, and pathotroph–saprotoph–symbiotrophs in the second year following disturbance (Figure 3).

Ordination revealed that the 2018 soil samples differed by soil location (i.e., bulk vs. rhizosphere soil) (Figure 1). Soil location explained 17.04% (*p* = 0.001) of the variation in the microbial community at Cedar Creek in 2018, and 4.85% (*p* = 0.007) of the variation in the community at Jena in 2018. The bacterial and fungal alpha diversity did not vary between soil locations. However, the bacterial alpha diversity by plant species differed at both sites, but fungal alpha diversity did not vary between plant species at either Cedar Creek or Jena (Appendix A).

Microbial composition differed by plant species and plant functional groups in the rhizosphere but not in the bulk soils (Table 3). The effects of plant functional groups on the microbial communities were driven by legumes: when legumes were removed from the dataset, the functional groups of the plants were no longer significant (R^2^ = 0.08 *p* = 0.052 for Cedar Creek; R^2^ = 0.07 *p* = 0.220 for Jena).

At both sites, *Ascomycota* were more abundant, and *Chytridiomycota*, *Mucoromycota*, and *Rozellomycota* were less abundant in the rhizosphere than bulk soils (Figure 4). At Cedar Creek, *Glomeromycota*, *Monoblepharomycota*, and *Mortierellomycota* had lower relative abundance in the rhizosphere than bulk soils. At Jena, *Kickxellomycota* had lower relative abundance in the rhizosphere than bulk soils. For fungal functional groups, pathotroph–symbiotrophs and symbiotroph–pathotroph–saprotrophs were more abundant in the rhizosphere than bulk soils at Cedar Creek. For bacteria, at Cedar Creek, *Firmicutes* had lower relative abundance and *Tenericutes* had higher relative abundance in the rhizosphere than bulk soils. At Jena, *Protobacteria* had lower relative abundance in the rhizosphere than bulk soils.

### 3.3. Relationship between Microbial Communities and Plant Biomass

The envfit function revealed that many microbial ASVs differed between sites, years, and soil locations (Appendix B). For the bulk and rhizosphere soils together, 233 fungal and 128 bacterial ASVs differed between the two sites (*p* < 0.05). For Cedar Creek, 38 fungal and 77 bacterial ASVs differed between years, and 23 fungal and 86 bacterial ASVs (*p* < 0.05) differed between soil locations. For Jena, eight fungal and seventy-two bacterial ASVs differed between years, and fourteen fungal and forty-eight bacterial ASVs (*p* < 0.05) differed between soil locations. For Cedar Creek, 55 ASVs differed among plant species. For Jena, 31 ASVs differed among plant species. PERMANOVA to assess the effect of plant species on individual phyla in the rhizosphere revealed that plant species significantly impact the composition of *Ascomycota* and *Basidiomycota* in both sites, *Glomeromycota* in Cedar Creek, and *Chytridiomycota*, *Mortierellomycota*, and *Zoopagomycota* in Jena (Table 4). Cedar Creek had no bacterial phyla significantly impacted, but, in Jena, the compositions of *Actinobacteria* and *Firmicutes* were significantly impacted by plant species.

To determine if NMDS score captures variation in microbial communities that cause differing plant responses (PSF), we regressed the plant biomass of a plot against the NMDS score for the microbial community of that same plot for 2018 rhizosphere data (Figure 5). NMDS axes 2 and 4 had significant relationships (Figure 5). There were 19 ASVs found in association with the low or high ends of NMDS2 (−0.3 > NMDS2 > 0.5, *p* < 0.05) and 22 ASVs found in association with the low or high ends of NMDS4 (−0.2 > NMDS4 > 0.3, *p* < 0.05). All these ASVs were bacteria. Of these ASVs, there were 12 in common between NMDS2 and NMDS4. NMDS2 and NMDS4 had opposite relationships to plant biomass (NMDS2 was positively correlated with plant biomass, while NMDS4 was negatively correlated with plant biomass). Four of the twelve ASVs in common between the two axes had similar relationships to plant biomass: *Bacteroides* spp. was positively correlated with logged plant biomass along both significant NMDS axes; *Blautia* spp., *Sellimonas* spp., and *Clostridium inocuum* were negatively correlated with logged plant biomass along both NMDS axes. The soils associated with both negative NMDS2 and positive NMDS4 (correlated with reduced plant biomass) were from KOEMAC, AMOCAN, and ANDGER, and the soils associated with positive NMDS2 and negative NMDS4 (correlated with increased plant biomass) were from LESCAP, LUPPER, SCHSCO, and SORNUT.

## 4. Discussion

With data from large field experiments on two continents, the results provide an unusually comprehensive perspective on microbial community composition in PSF experiments [30]. Sampling two experiments on two continents over four years allowed us to compare the magnitude of plant effects on soil microbial communities to the magnitude of site, year, and rhizosphere vs. bulk soil on soil microbial community composition. In descending order of importance, site, year, rhizosphere, and plant identity explained the variation in microbial community composition [58]. This result suggested that conditions associated with site and year, such as climate and soil type, have larger effects on soil microbial communities than short-term plant growth [31,59]. Not surprisingly, the effects of plant growth during the experiments (i.e., short-term plant effects) were greatest in rhizosphere soils [44]. Many studies have described this type of plant growth effect on the microbial community composition, but few studies have taken the next step of correlating these microbial community compositions with subsequent plant growth to identify putative plant-growth-suppressing or plant-growth-promoting soil organisms or communities of organisms [30].

In this study, much of the variation in rhizosphere microbial community composition associated with plant species was not correlated with subsequent plant growth (i.e., NMDS axes 1, 3, 5, and 6) [15]. This suggests that plants cause many changes in soil microbial community composition that do not feed back to affect plant growth. For example, it is likely that plants induce widespread changes in decomposer bacterial composition without causing large changes in decomposition rates. We did, however, find correlations between microbial community composition and plant growth for NMDS axes two and four at the Cedar Creek site. These correlations suggest that the organisms in these communities are likely to cause plant growth responses. We did not find evidence that these correlations were caused by one or even a few ASVs. Rather, these correlations were caused by dozens of ASVs. The fact that ‘treatment’ differences were caused by many ASVs suggests that PSF is caused by a diverse mixture of soil organisms. The fact that plant effects on microbial community composition were only detected after other effects were removed and that there was moderately large overlap in ASV composition among treatments suggest that PSF is driven by changes to dozens of the thousands of ASVs that differ among sites, years, and soil location (i.e., bulk vs. rhizosphere soil) [25,27,60]. In other words, PSF appeared to be caused by many weak changes in a small subset of the whole microbial community.

There is a history, particularly in agricultural systems, of trying to identify individual plant pathogens and symbionts [61,62,63]. Yet, our results suggest that it is large communities of soil organisms and not individual symbionts or pathogens that affect plant growth [25,26]. This perspective may help to explain why PSF values vary in different settings (e.g., greenhouse vs. field or one site vs. another site) [62,64]. More specifically, our results suggest that PSFs vary in different sites because they reflect the complex interactions between site conditions, priority effects, and complex interactions among dozens to hundreds of ASVs [62,64]. Our results suggest that management techniques that affect whole microbial communities and not just individual microbial species may be needed to affect plant growth. Whole-soil inoculations, rotation cropping, pH manipulation, and activated carbon addition would be examples of this type of management approach [26,27,63,65].

The study site explained the most variation in microbial community composition. This was not surprising because the two sites are on separate continents with different climates and soil types. Jena, the more nutrient- and carbon-rich site with a higher pH, had a higher relative abundance of the saprotrophic *Mortierellomycota* fungal phyla, while Cedar Creek was dominated by *Ascomycota* fungi, which can have multiple trophic modes [57]. The sites, while on different continents with different climatic regimes and soil histories, still shared 29% of their ASVs, and the relative abundances of bacteria were similar at both sites. Consistent with previous studies, *Firmicutes* and *Proteobacteria* dominated bulk soils, although *Actinobacteria* were less common than in previous studies [66]. Plant species effects were not detectable when data from both sites were analyzed together.

Within sites, ‘year’ explained the most variation in microbial community composition. Year effects overshadowed the effects of plant species. It is reasonable to expect that plants would exert cumulative effects on soil microbial community composition over time. However, differences among years did not appear to be directional in ordination space. This suggests that interannual variation in weather and not plant-driven effects explained the variation in the microbial communities from year to year [1,67].

After ‘site’ and ‘year’, soil location (bulk vs. rhizosphere) explained the most variation in soil microbial communities. ASV richness was lower in rhizosphere soils than in bulk soils, a result supported in other studies and consistent with the idea that root exudation results in high resource availability and competitive suppression among soil organisms in the rhizosphere relative to bulk soils [44,46]. Alternatively, it is also possible that lower richness reflects plant ‘selection’ for some organisms and ‘suppression’ of other organisms. The rhizosphere soils had higher relative abundance of *Ascomycota* and lower relative abundance of *Chytridiomycota* and *Mucoromycota* than the bulk soils. This corresponded to an increase in pathotrophs in the rhizosphere compared to bulk soils in Cedar Creek, further supporting the standard PSF theory that most plants have negative PSF because, as they grow, plants accumulate pathogens that negatively affect their growth. Although distinct, the bulk and rhizosphere soils shared roughly 50% of their ASVs. Lower species richness and large overlap in ASV composition are consistent with the idea that the rhizosphere microbiota is a subset of the bulk microbial community whose growth has been increased by a plant species [59,68].

Because plant root exudates and leaf litter can shift microbial community composition [3,69], it is commonly believed that the rhizosphere microbial community is more closely tied to plant growth and health [59,70]. Consistent with this idea, we found that plant species effects on microbial community composition were only detectable in rhizosphere soils when the effects of larger scale variation (i.e., site, year, and soil location) had been removed. The rhizosphere in both sites had an increase in *Ascomycota*, which are part of all fungal guilds, and a decrease in *Chytridiomycota*, *Mucoromycota*, and *Rozellomycota* [57]. The last three Phyla are mostly undefined in terms of function. However, they may be mainly pathotrophs or pathotroph–saprotrophs, which could indicate that the plants are encouraging the growth of microbes that are not pathogens in their rhizospheres.

A fundamental but unresolved question in PSF research is whether soil microbial effects on plant growth are caused by a ‘few large’ or ‘many small’ interactions. We found no evidence to support the ‘few large’ hypothesis. In other words, PSFs did not appear to be driven by a few important pathogens or symbionts. Instead, we found that differences in microbial communities were caused by dozens or hundreds of ASVs. This was true for differences among sites, years, bulk, and rhizosphere soils and among soils cultivated by different plants. For example, the envfit analysis indicated that dozens to hundreds of ASVs were associated with NMDS2 and NMDS4, the axes that were correlated with Phase II plant growth. Other studies have shown that, while specific microbes are able to benefit or harm plants, their effects are greatly influenced by the rest of the microbial community, and microbe–microbe interactions may play a large part in PSF [68]. Our results may help to explain why single ASV inoculations have failed to affect microbial community composition or plant growth [60,61,71].

While not being able to identify individual ASVs that drive PSF makes it more challenging to manage soil microbial communities to manipulate plant growth, these results suggest that more ecological approaches may be needed for leveraging microbes in managing plant communities [25,26,27]. As entire microbial communities are likely important for plant growth, using approaches that manipulate the entire microbial community, such as rotation cropping or whole-soil inoculations, may be needed [26,27,63]. For example, we found that three plant species (KOEMAC, AMOCAN, and ANDGER) from three different functional groups (C3 grasses, C4 grasses, and legume) appeared to create soil microbial communities that decreased subsequent plant growth. Similarly, four plant species (LESCAP, LUPPER, SCHSCO, and SORNUT) from two functional groups (legumes and C4 grasses) appeared to create soil microbial communities that increased plant biomass. Our results support several other studies that theorize that ubiquitous generalist microbes differentially affect plants and lead to changes in their growth [15,67,72]. The correlation between soil microbial communities and plant growth across several different plant species and functional groups suggests that it is these microbes, present in soils created by many different plants, that affect plant growth.

While our results suggest that diverse microbial communities determine plant growth responses, there were a few genera that may be candidates for future research on the microbial drivers of PSF; *Bacteroides* spp. were positively correlated with plant growth and may be a possible plant growth promoter, and *Blautia* spp., *Sellimonas* spp., and *Clostridium* inoculum were all negatively correlated with plant growth and may be possible plant pathogens. The ability to use multivariate methods to pinpoint specific microbial communities or ASVs that are correlated with plant growth is a promising avenue for further research into the mechanisms driving PSF, particularly in agricultural settings where less-diverse microbial communities may be more responsive to inoculations [72].

## 5. Conclusions

The variation in microbial communities was greatest where the long-term effects of soil type and climate can accumulate, but the shorter-term effects of weather and plant growth were also detectable.There was little evidence at any treatment level that differences associated with treatments were caused by a few soil organisms [60,72]. Instead, evidence across treatments suggested that large, diverse assemblages of soil organisms interact to define the soil microbial community structure and the effect on plant growth.The management implication for this finding is that whole-soil manipulations may be necessary to manage plant growth through soil microbial communities. Whole-soil inoculations, soil amendments such as biochar, compost, or activated carbon, and crop rotation may be more effective than single-organism inoculations [25,62,64].

## Figures and Tables

**Figure 1 biology-12-01374-f001:**
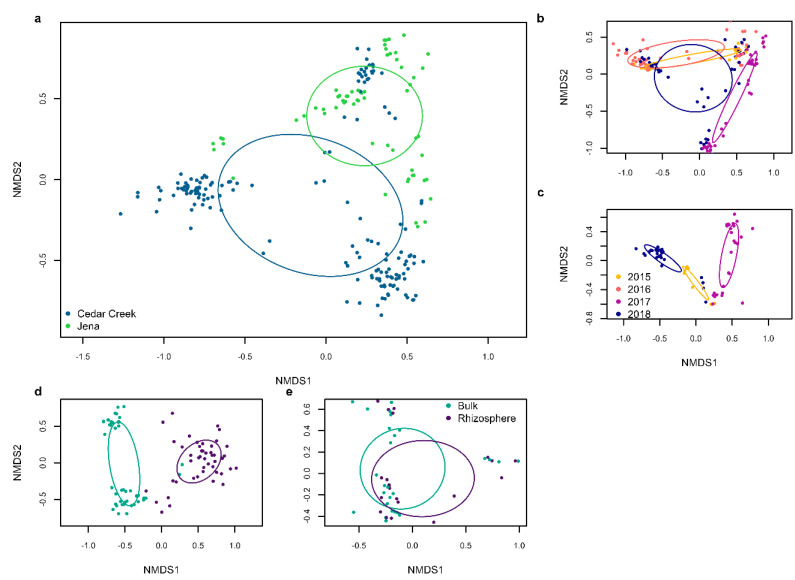
Non-metric multidimensional scaling (NMDS) of microbial communities throughout the study. (**a**) NMDS of bulk soil microbial communities from all years of the study at both Cedar Creek and Jena. (**b**) NMDS of bulk soil microbial communities at Cedar Creek for all years of the study. (**c**) NMDS of the bulk soil microbial communities at Jena for all years of the study. (**d**) NMDS of the 2018 data, including both bulk and rhizosphere data at Cedar Creek. (**e**) NMDS of the 2018 data, including both bulk and rhizosphere data at Jena.

**Figure 2 biology-12-01374-f002:**
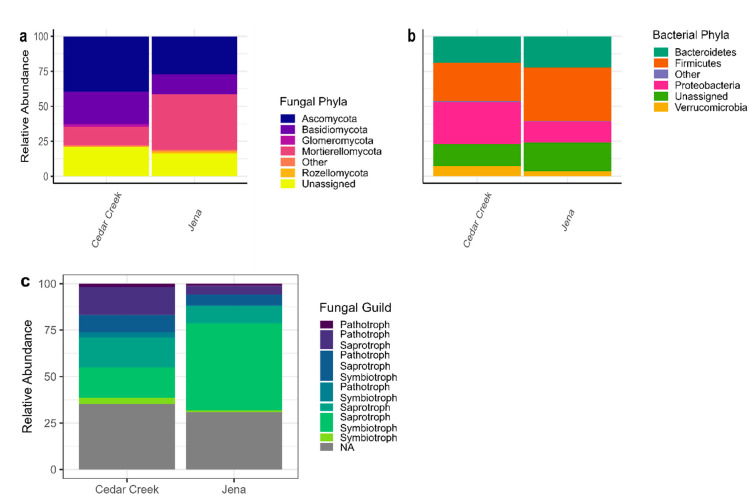
The relative abundance of the bulk soil microbes and their functional guilds at both sites. (**a**) The relative abundance of fungal phyla at both Cedar Creek and Jena. (**b**) The relative abundance of bacterial phyla at both Cedar Creek and Jena. (**c**) The relative abundance of the fungal guilds at Cedar Creek and Jena.

**Figure 3 biology-12-01374-f003:**
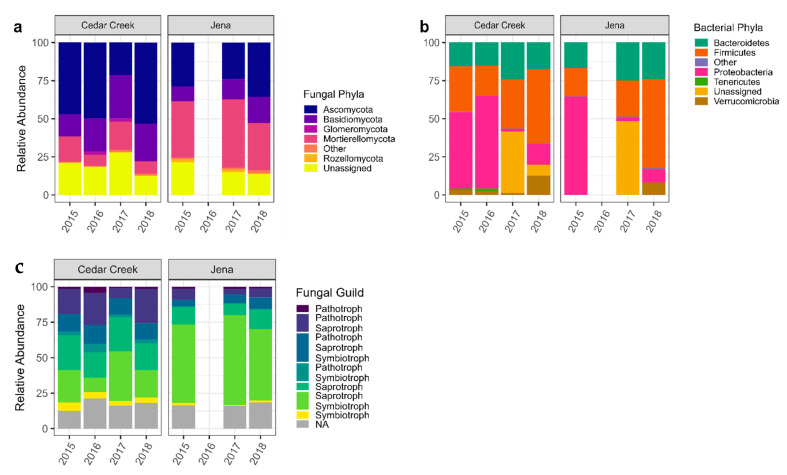
The relative abundance of the bulk soil microbes for each year of the study. (**a**) The relative abundance of the bulk soil fungal phyla throughout all years of the study at Cedar Creek and Jena. (**b**) The relative abundance of the bulk soil bacterial phyla throughout all years of the study at Cedar Creek and Jena. (**c**) The relative abundance of the fungal guilds for each year of the study at Cedar Creek and Jena.

**Figure 4 biology-12-01374-f004:**
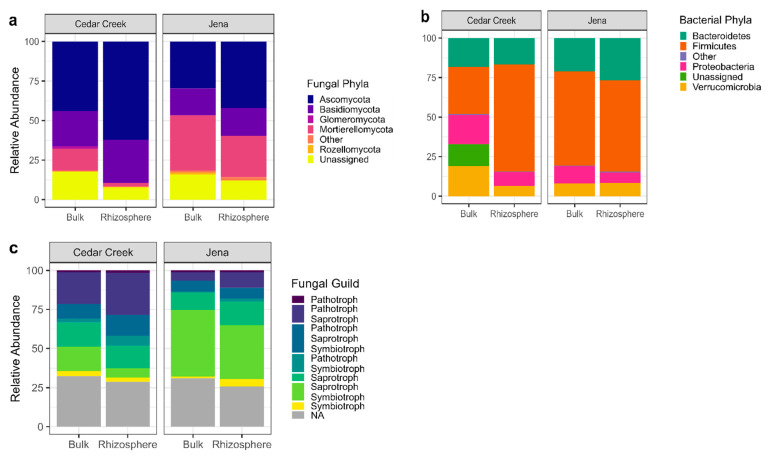
The relative abundance of the bulk and rhizosphere microbes in 2018. (**a**) The relative abundance of the fungal phyla for the bulk and rhizosphere soils at Cedar Creek and Jena. (**b**) The relative abundance of the bacterial phyla for the bulk and rhizosphere soils at Cedar Creek and Jena. (**c**) The relative abundance of the fungal guilds at Cedar Creek and Jena.

**Figure 5 biology-12-01374-f005:**
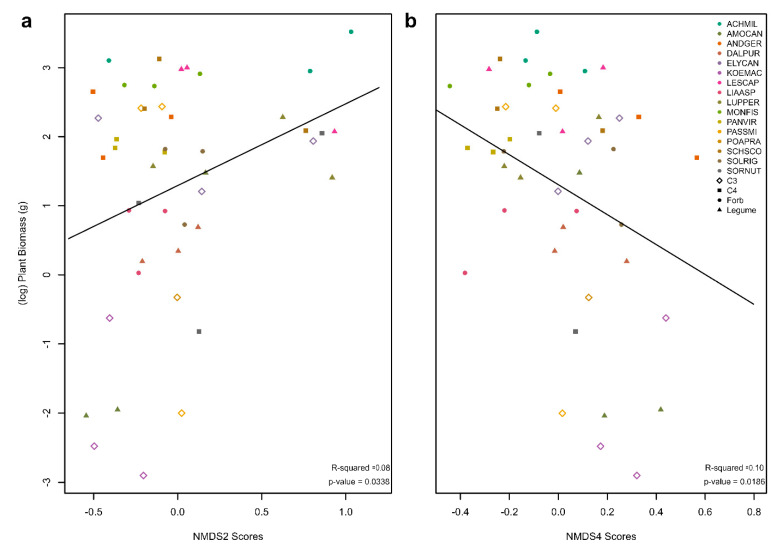
Regression of logged plant biomass against the NMDS score of the soil type that plant grew on. (**a**) Regression of logged plant biomass against NMDS2 at Cedar Creek. (**b**) Regression of logged plant biomass against NMDS4 at Cedar Creek. The plant biomass of a plot was regressed against the NMDS score for the microbial community of that same plot for 2018 rhizosphere data.

**Table 1 biology-12-01374-t001:** Plant species, functional groups, and species codes used at Cedar Creek.

Species	Functional Group	Code
*Amorpha canescens*	Legume	AMOCAN
*Andropogon gerardii*	C4	ANDGER
*Achillea millefolium*	Forb	ACHMIL
*Dalea purpurea*	Legume	DALPUR
*Elymus canadensis*	C3	ELYCAN
*Koeleria macrantha*	C3	KOEMAC
*Liatris aspera*	Forb	LIAASP
*Lespedeza capitata*	Legume	LESCAP
*Lupinus perennis*	Legume	LUPPER
*Monarda fistulosa*	Forb	MONFIS
*Poa pratensis*	C3	POAPRA
*Pascopyrum smithii*	C3	PASSMI
*Panicum virgatum*	C4	PANVIR
*Sorghastrum nutans*	C4	SORNUT
*Solidago rigida*	Forb	SOLRIG
*Schizachyrium scoparium*	C4	SCHSCO

**Table 2 biology-12-01374-t002:** Plant species, functional groups, and species codes used at Jena.

Species	Functional Group	Code
*Alopecurus pratensis*	C3	ALOPRA
*Anthirscus sylvestris*	Forb	ANTSYL
*Arrhenatherum elatius*	C3	ARRELA
*Dactylis glomerata*	C3	DACGLO
*Geranium pratense*	Forb	GERPRA
*Phleum pratense*	C3	PHLPRA
*Poa trivialis*	C3	POATRI
*Trifolium pratense*	Legume	TRIPRA
*Trifolium repense*	Legume	TRIREP

**Table 3 biology-12-01374-t003:** Summary of effects of plant species and functional group by soil location.

		Cedar Creek		Jena	
Independent Variable	Dependent	R^2^	*p*-Value	R^2^	*p*-Value
Plant species	Bulk soil	0.09	0.531	0.13	0.516
Plant species	Rhizosphere soil	0.40	0.002	0.48	0.004
Plant functional group	Bulk soil	0.02	0.329	0.03	0.430
Plant functional group	Rhizosphere soil	0.11	0.004	0.17	0.007

**Table 4 biology-12-01374-t004:** Summary of plant species’ effects on microbial phyla in rhizosphere. Results of a PERMANOVA testing the plant species effect on each microbial phylum.

Bacteria					Fungi				
	Cedar Creek		Jena			Cedar Creek		Jena	
Phylum	R^2^	*p*-Value	R^2^	*p*-Value	Phylum	R^2^	*p*-Value	R^2^	*p*-Value
*Actinobacteria*	0.67	0.166	**0.58 ***	**0.019**	*Ascomycota*	**0.49**	**0.001**	**0.49**	**0.001**
*Bacteroidetes*	0.38	0.792	0.67	0.369	*Basidiobolomycota*	NA	NA	0.47	0.9889
*Cyanobacteria*	NA **	NA	NA	NA	*Basidiomycota*	**0.41**	**0.001**	**0.41**	**0.016**
*Firmicutes*	0.33	0.411	0.46	0.150	*Calcarisporiellomycota*	NA	NA	NA	NA
*Patescibacteria*	NA	NA	NA	NA	*Chytridiomycota*	0.44	0.792	0.40	0.165
*Proteobacteria*	0.32	0.584	**0.49**	**0.013**	*Entorrhizomycota*	NA	NA	NA	NA
*Tenericutes*	NA	NA	NA	NA	*Glomeromycota*	**0.55**	**0.001**	**0.59**	**0.044**
*Verrucomicrobia*	0.42	0.543	NA	NA	*Kickxellomycota*	NA	NA	0.46	0.152
					*Monoblepharomycota*	NA	NA	NA	NA
					*Mortierellomycota*	**0.44**	**0.013**	**0.60**	**0.002**
					*Mucoromycota*	0.62	0.165	NA	NA
					*Neocallimastigomycota*	NA	NA	NA	NA
					*Olpidiomycota*	NA	NA	0.65	0.11
					*Rozellomycota*	NA	NA	0.38	0.269
					*Zoopagomycota*	NA	NA	**0.54**	**0.011**

* Significant values (α < 0.05) indicated in bold. ** For values with NA: either that phylum was not present in that site, or there were not enough samples with that phylum to conduct a statistical test.

## Data Availability

Sequence Read Archive (SRA) has accession number PRJNA683074 and is available at www.ncbi.nlm.nih.gov/bioproject/683074. Plant biomass data are available in the Utah State University Digital Commons at https://doi.org/10.26078/52k0-jr94.

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
