# Peer review of "Are Plant–Soil Feedbacks Caused by Many Weak Microbial Interactions?"

_biology, 2023, doi:10.3390/biology12111374_

Round 1
Reviewer 1 Report (Previous Reviewer 2)
Comments and Suggestions for Authors
The manuscript underwent significant enhancement during the authors' revisions and corrections. They conscientiously attended to the majority of the feedback and concerns. The present version is now well-prepared for publication.
Reviewer 2 Report (Previous Reviewer 3)
Comments and Suggestions for Authors
The work submitted for re-review was improved in every aspect. A fundamental change has been made in the introduction (which was inadequate), results and discussion. All my comments have been incorporated into the text and I am satisfied. There are minor technical errors in the text, such as typos, which should be removed.
This manuscript is a resubmission of an earlier submission. The following is a list of the peer review reports and author responses from that submission.
Round 1
Reviewer 1 Report
Comments and Suggestions for Authors
The manuscript entitled "Are Plant-Soil Feedbacks Caused by Many Weak Microbial Interactions?" has been revised according to the suggestions in the review. I appreciate all the improvements to the manuscript. In my opinion, the article can be accepted for publication in the journal "Biology".
Author Response
Thanks. It appears there are no edits requested from this reviewer.
Reviewer 2 Report
Comments and Suggestions for Authors
The current manuscript entitled “Are Plant-Soil Feedbacks Caused by Many Weak Microbial Interactions?” investigated soil microbial community composition in a field plant-soil feedbacks (PSF) experiments conducted in Minnesota, USA, and Jena, Germany using high-throughput sequencing. The variation in microbial community composition in the following descending order of significance: between the two study locations, across different years, between bulk and rhizosphere soils, and among rhizosphere soils associated with various plant species was observed.
Comments:
The design of the experiment and the analyses carried out are unclear and need more explanation.
How was suggested that plants cause many changes in soil microbial community composition that do not feedback to affect plant growth? Using the current design, it is difficult to address the plant-soil-microorganisms interactions ignoring the effect of soil disturbance after 2 years by removing plant covers or using herbicides. However, the current study aimed to investigate the effect of disturbance “Line 109-110”. But, there was no controls to be tested to determine the effect of that disturbance on microbial community structure.
Within the abstract: How “many weak” plant-microbe interactions were measured? Also, which your results suggested that PSF was difficult to induce by inoculation?
How Linear regression was performed to assess the effect of microbial community on plant biomass and plant PSF score? Does that analysis perform on each individual plant species? Where are the results of plant biomass? How PSF score was calculated? How plant biomass was estimated.
The study compared microbial community composition across two sites with markedly distinct physical, chemical, and biological characteristics. One site featured 16 plant species, while the other had only 9. Can you elaborate on how these sites were deemed comparable?
Line 380-381, How was concluded that “microbial inoculations often fail to change plant growth [57,59]” based on your results or the conclusion of these two references? I feel that authors ignored many other works that found that microbial inoculations positively changed plant growth and microbial community composition.
Author Response
Response to reviewer 2.
The current manuscript entitled “Are Plant-Soil Feedbacks Caused by Many Weak Microbial Interactions?” investigated soil microbial community composition in a field plant-soil feedbacks (PSF) experiments conducted in Minnesota, USA, and Jena, Germany using high-throughput sequencing. The variation in microbial community composition in the following descending order of significance: between the two study locations, across different years, between bulk and rhizosphere soils, and among rhizosphere soils associated with various plant species was observed.
Comments:
The design of the experiment and the analyses carried out are unclear and need more explanation.How was suggested that plants cause many changes in soil microbial community composition that do not feedback to affect plant growth?
Response: To clarify, we have added the following to the end of the Introduction: ‘Finally, and perhaps most importantly, we were able to test whether the changes plants caused in microbial community composition were correlated with plant biomass. Positive correlations would suggest that a plant created a soil microbial community with plant growth promoting organisms. Negative correlations would suggest that a plant created a soil microbial community with plant growth suppressing organisms. No correlations suggest that plants cause many changes in the soil microbial community that do not feedback to affect subsequent plant growth. For example, plants may cause changes in saprophytic bacterial communities, but these changes in community composition may not change soil nutrient cycling (i.e., function).‘
We have also added the following to the first paragraph of the Discussion:
In this study, much of the variation in rhizosphere microbial community composition associated with plant species was not correlated with subsequent plant growth. This suggests that plants cause many changes in soil microbial community composition that do not feedback to affect plant growth. For example, it is likely that plants induce widespread changes in decomposer bacterial composition without causing large changes in decomposition rates.
Using the current design, it is difficult to address the plant-soil-microorganisms interactions ignoring the effect of soil disturbance after 2 years by removing plant covers or using herbicides. However, the current study aimed to investigate the effect of disturbance “Line 109-110”. But, there was no controls to be tested to determine the effect of that disturbance on microbial community structure.
Response: Disturbance, certainly had an effect on the microbial community, but the disturbance was the same across all treatments (i.e., we controlled for the effect of disturbance). We did try to assess this affect by comparing microbial communities either one or two years following disturbance. This test was interesting and informative, but not central to the paper. Because it appears to cause confusion, we have removed this test.
Within the abstract: How “many weak” plant-microbe interactions were measured?
Response: We have tried to clarify this point as follows in the abstract: Two multivariate axes describing microbial community composition were correlated with plant growth. This suggested that the organisms important in these axes caused PSF. These multivariate axes weighted dozens of soil organisms suggesting that PSF was caused by ‘many weak’ plant-microbe interactions. Results suggest that PSFs may be difficult to reproduce or induce because they result from complex interactions that occur within the context of a much larger soil microbial community whose composition is determined by factors such as soil type, climate, weather, and priority effects.
Also, which your results suggested that PSF was difficult to induce by inoculation?
Response: We have removed (e.g., with inoculation). We did not test inoculations.
We agree with the reviewer that there are many examples of successful inoculation experiments. However, we also point out that a publication bias is likely to hide studies that did not find responses. More broadly, inoculations have been tested for decades, yet inoculations are not a standard agricultural practice suggesting that they are either not yet consistently effective or they are cost prohibitive.
How Linear regression was performed to assess the effect of microbial community on plant biomass and plant PSF score?
Response: We have added ‘(lm function) to clarify.
Does that analysis perform on each individual plant species?
Response: To clarify, we have rewritten this section as follows: ‘Linear regression (lm function) was performed to assess the effect of microbial community on plant biomass (log transformed). The logged biomass of a species in a plot and the NMDS score of the microbial community in that plot represent the y and x coordinates of that point. Regression was only performed for Cedar Creek data as the rhizosphere soil in Jena was sampled from different plots than the plant biomass data.
How plant biomass was estimated. Where are the results of plant biomass? How PSF score was calculated?
Response: Plant biomass was previously reported. We have added to the introduction that aboveground plant biomass was harvested for each plot. We are very clear in the introduction and methods that plant biomass values are reported elsewhere. It would be inappropriate to report these values again in this paper because they have already been published. We have removed reference to PSF scores.
The study compared microbial community composition across two sites with markedly distinct physical, chemical, and biological characteristics. One site featured 16 plant species, while the other had only 9. Can you elaborate on how these sites were deemed comparable?
Response:It is possible to compare the microbial communities in any two sites. Soil communities in tropical and arctic soils can be compared. We are not arguing that these sites are similar. We compare the microbial community composition among sites to provide a reference for the pattern and scale of differences among microbial communities in different sites, years, soil locations and plant growth histories. The idea is to provide context for any plant-induced changes. There are many microbial biogeography studies that similarly compare microbial communities in very different site conditions (e.g.,
Haiyan, C.; Gui-Feng, G.; Yuying, M.; Kunkun, F.; Manuel, D.-B. Soil Microbial Biogeography in a Changing World: Recent Advances and Future Perspectives. mSystems 2020, 5, 10.1128/msystems.00803-19, doi:10.1128/msystems.00803-19.
Fierer, N. Embracing the Unknown: Disentangling the Complexities of the Soil Microbiome. Nat Rev Microbiol 2017, 15, 579–590.
Fierer, N.; Jackson, R.B. The Diversity and Biogeography of Soil Bacterial Communities. Proceedings of the National Academy of Sciences 2006, 103, 626–631.
Line 380-381, How was concluded that “microbial inoculations often fail to change plant growth [57,59]” based on your results or the conclusion of these two references? I feel that authors ignored many other works that found that microbial inoculations positively changed plant growth and microbial community composition.
Response: We have removed this sentence. There are certainly many examples of inoculation affecting plant growth, particularly whole-soil inoculations, but given that the decades of efforts at developing inoculants, the fact that inoculants are not yet commonplace suggests either that inoculations remain ineffective or cost-prohibitive.
Reviewer 3 Report
Comments and Suggestions for Authors
The article's topic is interesting and attempts to find feedback on the plant-soil system.
The abstract is written in incomprehensible language to readers and lacks of numerical data and the names of species. Information on climate, soil types and weather should be removed from the abstract as research has yet to be carried out on this element.
Keywords: It is better to avoid repeating words in the title.
Introduction: Many papers have been devoted to this topic, e.g. Plant-soil Interactions: Ecological Aspects and Evolutionary Implications at various world ecoregions. It is a pity that the authors did not mention this in the introduction, but they should have.
Line 31- What does self' soils mean?
The objectives of the work should be defined at the end of the introduction and stated in bullet points to make them explicit.
Line 107-116. The fragment should be removed from here and moved to the results and discussion section.
Line 128- The name of the soil should be given according to the WRB, and briefly characterise its properties.
The results and discussion section is not fundamentally objectionable. As you can see, the work is already well-reworked after the initial revision.
The section of the Conclusion should be rewritten and given point forms. Actually, it's some summarising from the results.
Author Response
Response to reviewer 3.
The article's topic is interesting and attempts to find feedback on the plant-soil system.
The abstract is written in incomprehensible language to readers and lacks of numerical data and the names of species.
Response: We have rewritten to the abstract to try to make it clearer. We examined 25 plant species, so it is not possible to list the relevant plant species or thousands of soil organisms.
Information on climate, soil types and weather should be removed from the abstract as research has yet to be carried out on this element.
Response: We have rewritten as follows to address this comment: ‘Results suggest that PSFs may be difficult to reproduce or induce because they result from complex interactions that occur within the context of a much larger soil microbial community whose composition is determined by factors associated with site differences (e.g., soil pH, soil type, climate, weather, and priority effects).’
Keywords: It is better to avoid repeating words in the title.
Response: Thanks, we have removed repeated words.
Introduction: Many papers have been devoted to this topic, e.g. Plant-soil Interactions: Ecological Aspects and Evolutionary Implications at various world ecoregions. It is a pity that the authors did not mention this in the introduction, but they should have.
Response: We have added several sentences and citations to the introduction regarding previous research on the biogeography of soil microbial communities and their relation to plant-soil feedbacks.
More specifically, we have added the following references:
Haiyan, C.; Gui-Feng, G.; Yuying, M.; Kunkun, F.; Manuel, D.-B. Soil Microbial Biogeography in a Changing World: Recent Advances and Future Perspectives. mSystems 2020, 5, 10.1128/msystems.00803-19, doi:10.1128/msystems.00803-19.
Fierer, N. Embracing the Unknown: Disentangling the Complexities of the Soil Microbiome. Nat Rev Microbiol 2017, 15, 579–590.
Bauer, J.T.; Blumenthal, N.; Miller, A.J.; Ferguson, J.K.; Reynolds, H.L. Effects of Between-Site Variation in Soil Microbial Communities and Plant-Soil Feedbacks on the Productivity and Composition of Plant Communities. Journal of Applied Ecology 2017, 54, 1028–1039, doi:https://doi.org/10.1111/1365-2664.12937.
Ke, P.-J.; Miki, T.; Ding, T.-S. The Soil Microbial Community Predicts the Importance of Plant Traits in Plant–Soil Feedback. New Phytologist 2015, 206, 329–341, doi:https://doi.org/10.1111/nph.13215.
Fierer, N.; Jackson, R.B. The Diversity and Biogeography of Soil Bacterial Communities. Proceedings of the National Academy of Sciences 2006, 103, 626–631.
Schlatter, D.C.; Bakker, M.G.; Bradeen, J.M.; Kinkel, L.L. Plant Community Richness and Microbial Interactions Structure Bacterial Communities in Soil. Ecology 2015, 96, 134–142.
Herrera Paredes, S.; Lebeis, S.L. Giving Back to the Community: Microbial Mechanisms of Plant–Soil Interactions. Funct Ecol 2016, 30, 1043–1052, doi:https://doi.org/10.1111/1365-2435.12684.
Line 31- What does self' soils mean?
Response: We have clarified this as follows: ‘then in Phase II, each species is grown on ‘self’ or ‘other’ soils (i.e., ‘self’ soils are cultivated by the same species and ‘other’ soils arecultivated by different plant species)’
The objectives of the work should be defined at the end of the introduction and stated in bullet points to make them explicit.
Response: We have added two objectives to the end of the introduction.
Line 107-116. The fragment should be removed from here and moved to the results and discussion section.
Response: We have moved this sentence to the Discussion.
Line 128- The name of the soil should be given according to the WRB, and briefly characterise its properties.
Response: We have added the WRB soil name and pH and soil carbon values for the two sites.
The results and discussion section is not fundamentally objectionable. As you can see, the work is already well-reworked after the initial revision.
The section of the Conclusion should be rewritten and given point forms. Actually, it's some summarising from the results.
Response: We have reformatted the conclusions to three bullet points.